# Identification of *BRCA1/2* mutation female carriers using circulating microRNA profiles

Kevin Elias[1,11], Urszula Smyczynska [2,11], Konrad Stawiski[2], Zuzanna Nowicka [2], James Webber[1], Jakub Kaplan[3], Charles Landen[4], Jan Lubinski[5], Asima Mukhopadhyay[6], Dona Chakraborty[6], Denise C. Connolly[7], Heather Symecko[8], Susan M. Domchek [8], Judy E. Garber[9,10], Panagiotis Konstantinopoulos [9,10], Wojciech Fendler [2,3] ✉ & Dipanjan Chowdhury [3,9,10] ✉

Identifying germline *BRCA1/2* mutation carriers is vital for reducing their risk of breast and ovarian cancer. To derive a serum miRNA-based diagnostic test we used samples from 653 healthy women from six international cohorts, including 350 (53.6%) with *BRCA1/2* mutations and 303 (46.4%) *BRCA1/2* wild-type. All individuals were cancer-free before and at least 12 months after sampling. RNA-sequencing followed by differential expression analysis identified 19 miRNAs significantly associated with *BRCA* mutations, 10 of which were ultimately used for classification: hsa-miR-20b-5p, hsa-miR-19b-3p, hsa-let-7b-5p, hsa-miR-320b, hsa-miR-139-3p, hsa-miR-30d-5p, hsa-miR-17-5p, hsa-miR-182-5p, hsa-miR-421, hsa-miR-375-3p. The final logistic regression model achieved area under the receiver operating characteristic curve 0.89 (95% CI: 0.87–0.93), 93.88% sensitivity and 80.72% specificity in an independent validation cohort. Mutated gene, menopausal status or having preemptive oophorectomy did not affect classification performance. Circulating microRNAs may be used to identify *BRCA1/2* mutations in patients of high risk of cancer, offering an opportunity to reduce screening costs.

Hereditary breast and ovarian cancer (HBOC) is the most common hereditary cancer syndrome, and the two most commonly mutated genes in HBOC, *BRCA1* and *BRCA2*, both play critical roles in mediating DNA repair through homologous recombination (HR)[1]. Germline mutations in *BRCA1/2* account for 10–15% of ovarian cancers, 5–10% of breast cancers, and 3–5% of pancreatic and prostate cancers[2–7]. Loss of HR, known as HR deficiency (HRD), impairs the ability of cells to repair double-strand DNA breaks, leaving cells vulnerable to mutagenesis

from ionizing radiation and oxidative stress[8]. Identification of *BRCA1/2* mutation carriers is an essential component of cancer risk-reduction strategies and presents opportunities for cascade testing of other family members[9]. Mutation carriers have several opportunities for cancer prevention or interception, including risk-reducing salpingo-oophorectomy or mastectomy, hormonal chemoprevention, and enhanced surveillance protocols, such as MRI-based breast cancer screening[10–16].

[1]Division of Gynecologic Oncology, Brigham and Women's Hospital, Boston, MA, USA. [2]Department of Biostatistics and Translational Medicine, Medical University of Lodz, Lodz, Poland. [3]Department of Radiation Oncology, Dana-Farber Cancer Institute, Boston, MA, USA. [4]Department of Obstetrics and Gynecology, University of Virginia, Charlottesville, VA, USA. [5]International Hereditary Cancer Center of the Pomeranian Medical University, Szczecin, Poland. [6]Kolkata Gynecology Oncology Trials and Translational Research Group, Kolkata, West Bengal, India. [7]Fox Chase Cancer Center, Philadelphia, PA, USA. [8]Basser Center for BRCA, University of Pennsylvania, Philadelphia, PA, USA. [9]Center for BRCA and Related Genes, Dana-Farber Cancer Institute, Boston, MA, USA. [10]Harvard Medical School, Boston, MA, USA. [11]These authors contributed equally: Kevin Elias, Urszula Smyczynska. ✉e-mail: Wojciech.fendler@umed.lodz.pl; Dipanjan_Chowdhury@dfci.harvard.edu

Prevention or early detection of *BRCA1/2*-related cancers is predicated on the identification of *BRCA1/2* mutation carriers. At present, genetic testing for *BRCA1/2* is only recommended for individuals with a known personal or familial history of breast, ovarian, tubal, or primary peritoneal cancer or for persons descending from populations with high mutational prevalence (e.g., Ashkenazi Jewish)[17]. However, more than half of all carriers with *BRCA1/2* mutations have no family history of cancer, which would prompt a referral for genetic testing[18]. Among the estimated 1 million *BRCA1/2* mutation carriers in the United States, only 10% are aware of their carrier status[19].

While universal genetic testing might not be feasible or desirable, a functional screen for "*BRCA*ness" could improve the efficiency of cancer early detection and prevention efforts. Such a test could focus genetic counseling and testing among those individuals with the highest pretest probability of having a pathogenic mutation, regardless of personal or family history. We suggest microRNAs (miRNAs) might play a role in developing such a tool. Our teams and others have shown that miRNAs are directly linked to *BRCA*-mediated DNA repair[20–24]. HBOC-related tumors are characterized by distinct miRNA profiles from sporadic disease[25–29]. Furthermore, miRNAs circulate in blood, and circulating miRNAs are characterized by surprising stability and reproducibility, making them attractive circulating biomakers[30,31]. Previously, using sera from subjects with unknown *BRCA* status, we reported and validated a test based on circulating miRNA that produced both high positive and negative predictive values for discriminating ovarian cancers from benign pelvic masses[32]. Other groups subsequently reported similar findings[33,34]. We undertook the present study to investigate whether circulating miRNAs might vary by *BRCA1/2* mutational status. We hypothesized that circulating miRNAs profiles could be used to identify germline *BRCA1/2* mutations among otherwise healthy individuals without cancer.

In this work, we show that a panel of miRNAs can be used to identify *BRCA1/2* mutation carriers among healthy women with high genetic risk of ovarian or breast cancer. The serum miRNA-based test may provide a cheap first-line screening, guiding further efforts for genetic counseling and improving cancer prevention and early detection.

## Results

### Characteristics of the study population

The study population characteristics are summarized in Table 1. In total, samples were collected from 653 study subjects from six separate cohorts (Fig. 1). Among the study population, 350 (53.6%) subjects had *BRCA1* or *BRCA2* mutations (*BRCA*-mt), and 303 (46.4%) were *BRCA1/2*−wild-type (*BRCA*-wt). Summary clinical characteristics of each group are presented in Supplementary Table 1. A small number of participants (75/653; 11.5%) had undergone risk-reducing salpingo-

oophorectomy prior to blood collection because of *BRCA*-associated cancer risk, which was accounted for in the differential expression analysis.

### Identification of miRNAs associated with germline BRCA mutations

Unsupervised, linear and non-linear dimensionality reduction with PCA and UMAP were used to examine the effects of *BRCA1/2* deficient status and that of the batch (Fig. 2a, b and Supplementary Fig. 1). The batch effect clearly separated the groups. However, within both observed batches, the *BRCA* status strongly affected expression profiles (Fig. 2a). We aimed to identify differentially expressed (DE) miRNAs according to germline *BRCA1/2* mutations by superimposing the results after two strategies of data preprocessing−on raw data (Fig. 2c and Supplementary Data 1) and after batch adjustment (Fig. 2d and Supplementary Data 2). Nineteen miRNAs were convergent (*P* < 0.01 with |log$_2$(FC)| > 0.5 in the same direction in both variants, ratio of FCs from two analysis variants between 0.8 and 1.25) regardless of the data preprocessing strategy (purple markings in Fig. 2c, d). Unsupervised hierarchical clustering of all subject samples from the 5 groups used for miRNA selection and model development showed that the samples clustered based on the *BRCA1/2* mutations (Fig. 2e) with no evident preference towards *BRCA1* or *BRCA2* mutations. Notably, in the validation group composed of UPenn samples, the 19 miRNAs also clearly separated *BRCA*-mt and *BRCA*-wt samples confirming the robustness of their selection (Supplementary Fig. 2).

### Using miRNAs to predict BRCA mutation status

Having preselected 19 miRNAs with consistent capability of separating *BRCA*-mt from *BRCA*-wt samples, we used OmicSelector-based development of models to differentiate between *BRCA*-mt and wild-type samples based on batch-adjusted log2 (TPM) expression values. Feature sets derived from the training set (Supplementary Data 3) were used for modeling using four different approaches. The best predictive performance was achieved by a logistic regression model with parameters shown in Supplementary Table 2 based on 10 miRNAs: hsa-miR-20b-5p, hsa-miR-19b-3p, hsa-let-7b-5p, hsa-miR-320b, hsa-miR-139-3p, hsa-miR-30d-5p, hsa-miR-17-5p, hsa-miR-182-5p, hsa-miR-421, and hsa-miR-375-3p (Supplementary Data 4). This set of miRNAs was selected using feature ranking based on ROC AUC and a minimal description length (MDL) discretization algorithm on the training set balanced with the Synthetic Minority Oversampling Technique (SMOTE)[35].

The final model achieved 82.35% accuracy, 84.51% sensitivity, and 79.39% specificity on the original training set. The training AUC ROC (Fig. 3a) was 0.89 (95% CI: 0.87−0.93). This model achieved 84.62% accuracy, 95.33% sensitivity, and 83.64% specificity on the testing set and 85.61% accuracy, 93.88% sensitivity, and 80.72% specificity in the external validation set comprised of the UPenn group. Confusion matrices for the separate sets are available in Supplementary Table 3. Predicted probabilities of *BRCA*-mt in the context of true *BRCA* status are presented in Fig. 3b. Menopausal status (Fig. 3c and Supplementary Table 4) or having preemptive oophorectomy before blood sample draw (Fig. 3d) did not affect classification performance. Case-wise prediction for all available samples with clinical data and miRNA expression for all 19 miRNAs are presented in Supplementary Data 5. The presented diagnostic performance was calculated for the cutoff established on the basis of optimal accuracy. However, to better evaluate the utility of the proposed test we present the estimated positive and negative predictive values of different thresholds (based on the results from the whole patient cohort) for populations of varying prevalence of mutations in genes associated with homologous recombination pathway of DNA repair (Supplementary Fig. 3). Although data on accurate age at testing was provided for 52% of subjects, with a predominance of controls, we did not observe any

### Table 1 | Clinical characteristics of the studied group

| Variable | | Subjects (N = 653) |
|---|---|---|
| Menopausal status | Post-menopausal | 154 (23.6%) |
| | Pre-menopausal | 187 (28.6%) |
| | No data | 312 (47.8%) |
| Age, median (IQR), years | | 49 (39.2–60) |
| | Unknown | 313 (47.6%) |
| CA-125, median (IQR), IU | | 15 (10.3–31.2) |
| | Unknown | 552 (84.5%) |
| Having ovaries at testing | Yes | 578 (88.5%) |
| | No | 75 (11.5%) |

*IQR* interquartile range.

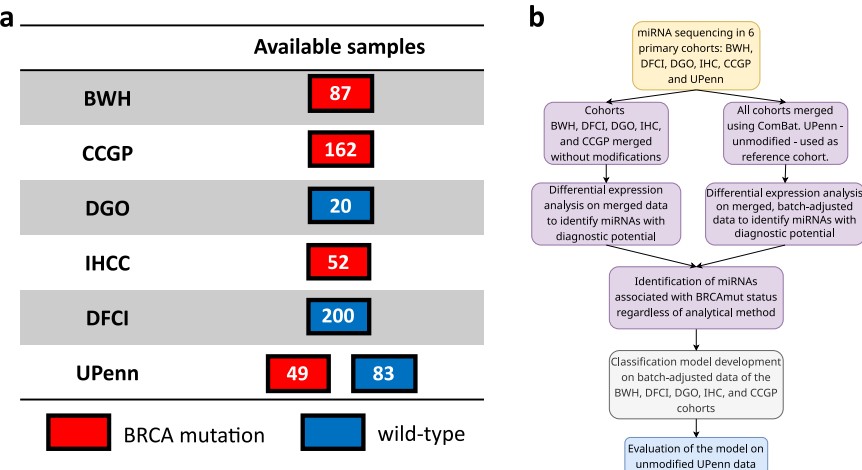

**Fig. 1 | miRNA expression data from healthy subjects with known germline BRCA1/2 mutation status. a** Overview of datasets. BWH Brigham and Women's Hospital, CCGP Center for Cancer Genetics and Prevention at DFCI, DGO Department of Gynaecological Oncology, Tata Medical Center, Kolkata, India, IHCC International Hereditary Cancer Center of the Pomeranian Medical University, Poland, DFCI DFCI/BWH biobank, UPenn University of Pennsylvania. **b** Scheme of statistical analysis design.

correlation between the model's predicted probabilities in both *BRCA*-mt ($r = 0.14$; $P = 0.33$) and *BRCA*-wt ($r = 0.01$; $P = 0.88$) individuals (Supplementary Fig. 4). The model's performance remained constant throughout the whole range of age categories (Supplementary Table 5).

## Discussion

In this study, we analyzed serum profiles of miRNA expression of a large ($N = 653$) group of healthy participants from six international cohorts to obtain a signature associated with *BRCA1/2* mutations. This is a clinically relevant finding because these individuals are at increased lifetime risk of developing *BRCA*-deficiency-related cancers. We used RNA sequencing for unbiased miRNA quantification and developed classification models to discriminate samples from subjects with *BRCA* mutations ($N = 350$) from those who are *BRCA* wild-type ($N = 303$). This work is distinct from previous studies which have either evaluated biomarker performance of circulating miRNAs directed at cancer diagnosis[33], focused on differences in miRNA-based *BRCA1/2* mutation signatures in the context of hereditary breast and ovarian cancers, or limited analyses to expression measured in formalin-fixed paraffin-embedded (FFPE) tumor tissues[29,36]. The present study is therefore a large-scale, comprehensive analysis of circulating miRNAs in healthy patients to identify those likely at high risk of hereditary cancers.

The presented test may serve as a balance to the United States Preventative Service Task Force recommendation against risk assessment, genetic counseling, or genetic testing for women "whose family history is not associated with an increased risk for harmful mutations in the *BRCA1/2* genes[17]." The argument to restrict testing derives from estimates that pathogenic mutations in *BRCA1/2* only occur in 0.2–0.3% of women in the general population[37] and a negative test result offers no gain in life expectancy nor eliminates the need for regular mammograms[38]. Despite falling costs for genetic testing, a cost-effectiveness investigation found that universal testing for the general population remains cost-prohibitive at about $1 million USD per quality-adjusted life year gained[38]. On the other hand, among patients referred for genetic testing based on family or personal cancer history, *BRCA1/2* mutations are identified in up to 25%[39]. The application of the miRNA-based test to identify patients at the highest risk offers an opportunity to reduce the costs of screening, which is particularly important in resource-limited settings.

Mechanistically, it has been shown by us[20] and other groups[40] that miRNAs regulate expression of DNA repair genes and may impact DNA repair capacity and sensitivity to poly (ADP-ribose) polymerase inhibitors (PARPi). The well-established dysregulation of miRNA expression in cancer, together with the contribution of miRNAs to tumorigenesis and the fact that in *BRCA1/2* mutation carriers, the genetic alterations are present in all body cells, offers a probable explanation for a distinct circulating miRNA signature. Haploinsufficiency of *BRCA1* or *BRCA2* gene for the suppression of replication stress instigated by environmental and endogenous factors was demonstrated, despite different biological functions of their encoded proteins[41–43]. This is supported by a recent observation of increased levels of soluble EGFR and increased thymidine kinase 1 activity in the sera of mutation carriers of either *BRCA1* or *BRCA2*[44]. Whether alterations in the levels of these miRNAs are an adaptive response to genomic instability or they function as messengers between cells remains to be established in future studies.

Homologous recombination haploinsufficiency is characterized by increased risks of ovarian, breast, pancreatic and prostate cancer as well as sensitivity to DNA damaging agents and PARPi. Although this phenotype, broadly termed *BRCA*ness, is most commonly associated with germline mutations in *BRCA1/2*, evidence from basic and clinical studies suggest that other genetic and epigenetic alterations may have similar effects on cancer risk, tumor molecular features, and drug sensitivity[45]. It is possible that a *BRCA*ness assay could help identify high-risk individuals with functional equivalency to *BRCA1/2* mutations who would not be identified by routine genetic tests, such as those with loss of *BRCA1/2* function through large-scale genomic rearrangements, promoter methylation, or mutations in less commonly mutated genes also in the HR repair pathway[46]. The matter of deploying the proposed test in clinical practice would also need to consider calibrating its cutoff to specific needs of the tested population. For the general population with the prevalence of germline *BRCA1/2* mutations in the range of ~0.4%, the negative predictive value (NPV) corresponding to a cutoff probability of 0.25 would be 99.6% and the positive predictive value (PPV) would be 8.7%, In other words, in a hypothetical population of 10,000 patients with low risk of mutations, our test with cutoff p set at 0.25 with 94.3% sensitivity and 58.1% specificity would correctly identify 38 of 40 patients with germline *BRCA1/2* mutations, while sparing 5787 of 9960 patients without mutations from costly genetic testing. For patients with higher risk of such variants—10.7% of mutations in HR genes that is observed in populations of patients diagnosed with breast cancer[47]—the same cutoff would yield an NPV of 98.8% and PPV of 21.2%. Individual-level decisions on

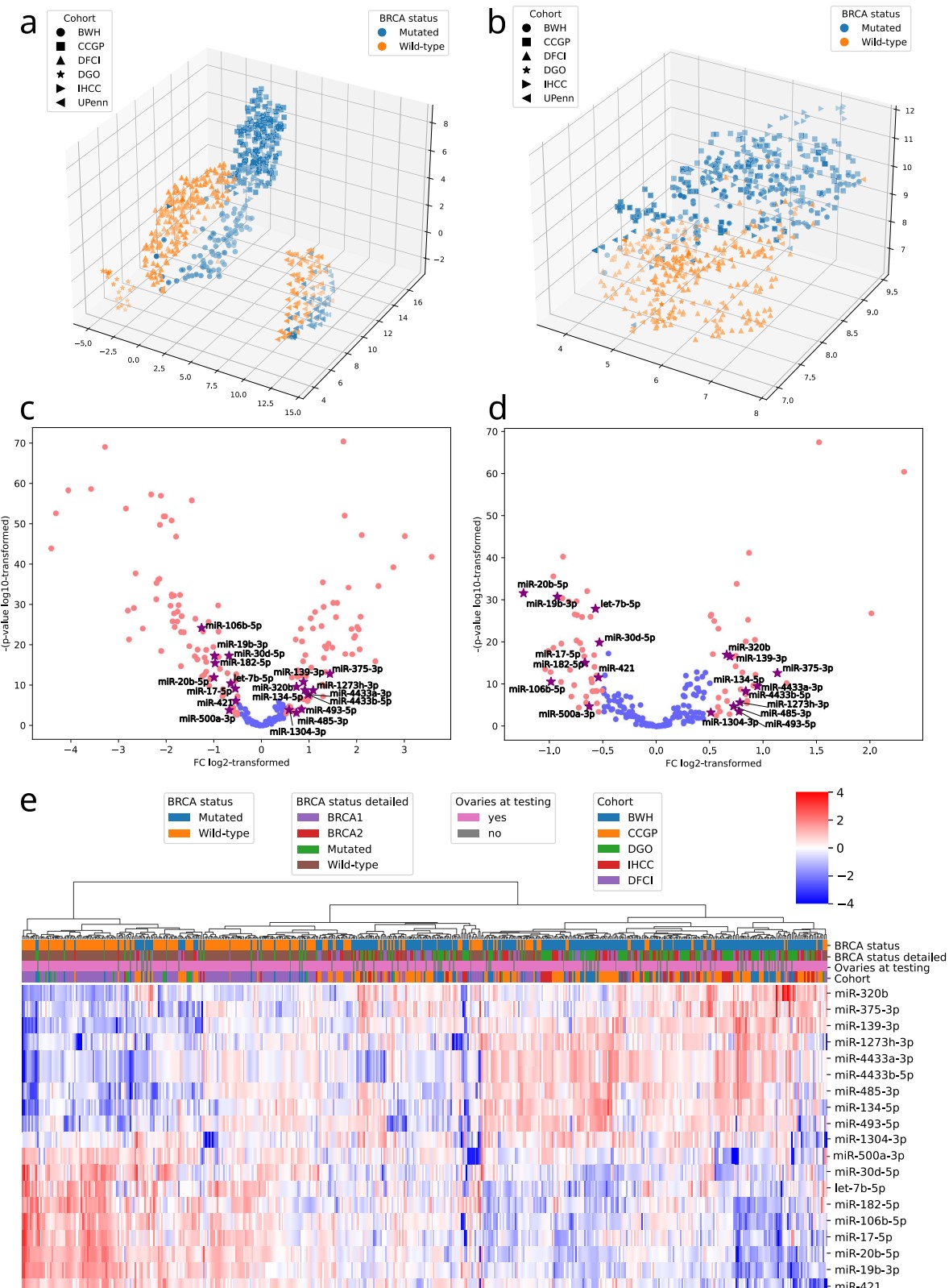

**Fig. 2 | Development of the *BRCA*-mt signature.** UMAP representation of *BRCA*-mt and *BRCA*-wt samples from all evaluated cohorts without (**a**) and after (**b**) batch adjustment (*N* = 653). Volcano plots showing differentially expressed miRNAs between *BRCA*-mutated and wild-type samples without (**c**) and after (**d**) batch adjustment (*N* = 521); red markings represent miRNAs with *P* < 0.01 and FC > 1.5 or <0.66; purple markings denote ones that were significant in both comparisons.

Limma package was used for between-group miRNA expression comparison, presented unadjusted *P* values, and FCs were calculated by limma algorithm.
**e** Heatmap of 19 miRNAs with convergent *BRCA*-mt; *BRCA*-wt profiles regardless of data preprocessing. Clustering primarily by *BRCA* status with no clear pattern of *BRCA*1 or *BRCA*2 or interference by prior oophorectomy (*N* = 521). Euclidean distance and complete linkage were used to determine cluster structure.

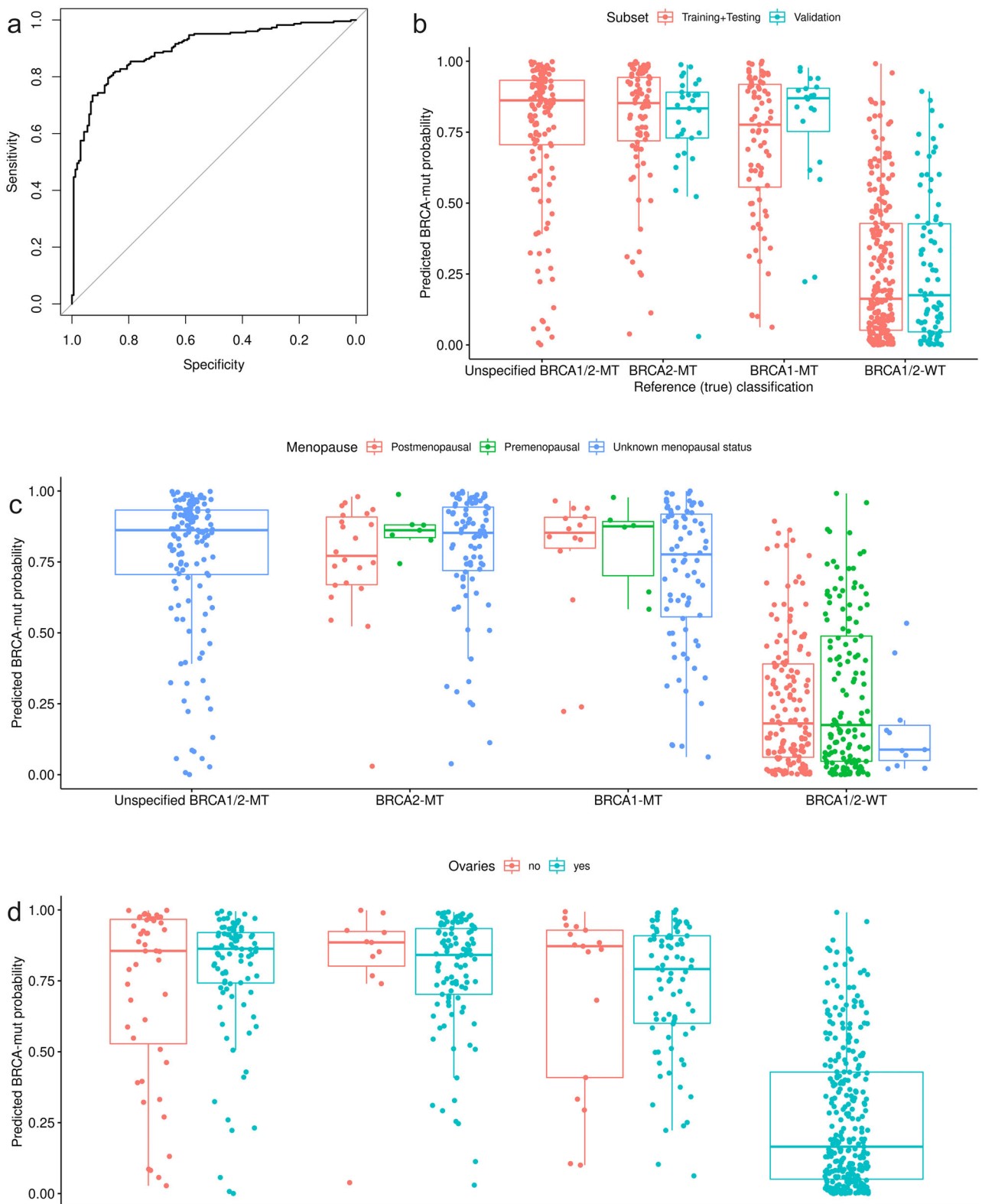

**Fig. 3 | Performance and *BRCA*1/2 mutation probability estimated by the final logistic regression model. a** Training (sample size *N* = 391) ROC curve of the final logistic regression model; the area under the curve equaled 0.89 (95% CI: 0.87–0.93); **b** predicted *BRCA1/2* mutation probability in training, testing, and validation sets according to reference mutational status; **c** predicted *BRCA*1/2 mutation probability in the context of menopausal status; **d** predicted *BRCA1/2* mutation probability in the context of having ovaries at the time of testing. **c**, **d** show that both menopausal status and lack of ovaries did not influence predicted *BRCA1* or *BRCA2* mutation probability. In boxplots median is marked as central line, boxes indicate the first and third quartile and whiskers present 1.5× IQR. **b**–**d** present all *N* = 653 samples, statistics were derived using all samples in respective subgroups.

using the test in prioritizing patients for genetic testing, however, could be made on the basis of the clinician's experience, patient's preference, available resources and screening programs. Currently, existing tests beyond germline mutation analysis, including 'genomic scar' assays, tissue transcriptional profiles, and functional HRD assays such as RAD51 foci quantification[48], fail to consistently identify these patient populations of interest. A test based on circulating miRNA expression has two important advantages: it is feasible to perform in the clinical setting without the need for tissue biopsy and it may provide a dynamic readout of the HR proficiency status[49]. We hope to test this in future work.

The current study has several important strengths. First, the miRNA test approach enriches a population at the highest risk for ovarian and other *BRCA* haploinsufficiency-related cancers. Individuals with a positive screen would still require genetic testing to confirm predisposition to cancer, thus reducing risks from false positive results. Likewise, patients screening negative but with a strong family history could still opt for genetic testing, minimizing the number of false negative results. The availability of a confirmatory test minimizes harm while ensuring that most at-risk patients are identified. Second, the resulting model, a logistic regression based on the expression of ten miRNAs, yielded 85.6% accuracy in the hold-out validation set, which we consider to be acceptable performance for an assay designed as a first-pass screen. However, as the data presented at this stage are generated through high throughput sequencing which would likely have to be downscaled to a simpler and cheaper method, the key issue at this stage was not to prioritize the performance of the model itself, but rather to identify variables with the best potential for class separation through any possible means. Additionally, the relatively uncomplicated structure of this model facilitates its explanation, avoiding the unexplainability problem suffered by more complex machine learning and artificial intelligence approaches. Third, the model was developed using data from 6 distinct groups from three continents covering different ethnicities and mutations both in *BRCA1* and *BRCA2*, which represents a more diverse group than our prior study and increases the likelihood that the results are generalizable. The wide range of study subject profiles further ensures that model performance was not affected by menopausal status or previously performed preventive surgery.

We also acknowledge the study's weaknesses. First, the presented models rely on next-generation sequencing (NGS). While the costs of this tool are coming down, and the use of NGS has entered some clinical applications (e.g., cell-free DNA for prenatal testing), other platforms, such as qRT-PCR, might be more efficient. Considerable batch effects related mainly to the application of different sequencing platforms is an issue that cannot be ignored as a potential obstacle in the translation of our results. Although its impact was largely mitigated by selection of miRNAs consistently dysregulated regardless of using batch correction, similar problems may arise in the future with application of other sequencing platforms and reagents. Finally, we have not investigated these models in patients with other types of DNA repair defects, such as Lynch Syndrome, nor conducted sensitivity analyses across various racial and ethnic subpopulations. Indeed, metadata (age, menopausal status) were not available for a large number of samples. While we did not see variations in model performance across these subgroups, we cannot completely exclude the possibility that these factors may be confounders in a larger sample size. These studies will be needed to examine the generalizability of the approach. Finally, BRCA1 and BRCA2 have distinct functions in the HR repair pathway, yet the haploinsufficiency of either gene gives us a common circulating miRNA signature. We speculate that haploinsufficiency in HR-mediated repair and consequent genomic instability is a potential cause of this miRNA-based signal in serum. However, we need to validate this idea with serum miRNA analysis from individuals with HR gene haploinsufficiency caused by genetic factors other than mutations of *BRCA1* or *BRCA2*.

In summary, we show that circulating miRNA levels can be used to stratify individuals as likely or unlikely to harbor a *BRCA1/2* mutation. This extends our prior finding that a diagnostic circulating miRNA model can help distinguish ovarian cancer cases from benign adnexal masses or controls[32]. The approach supplements our previous effort by providing a means to identify individuals at elevated risk for ovarian cancer who require careful monitoring and may be advised to undergo risk-reduction surgery. The result raises the potential for directing identified patients to serial assessment of ovarian cancer risk designed for high-risk populations, an approach under assessment in a nationwide prospective observational study known as the microRNA Detection (MiDe) Study (www.midestudy.org).

## Methods

The study was approved by the following ethical committees: Tata Medical Center−Institutional Review Board (approval number 2018/TMC/117/IRB6), Institutional Review Board of Dana-Farber Cancer Institute (#13−325), Institutional Review Board of University of Pennsylvania (#816688), Ethics Committee of the Pomeranian Medical University in Szczecin (BN-001/174/05).

### Samples

The study group was assembled from six serum biorepositories (Fig. 1a) based at: Brigham and Women's Hospital (BWH; Boston, MA; $N = 87$), Dana-Farber Cancer Institute (DFCI; Boston, MA; $N = 200$), a separate sample set from the Center for Cancer Genetics and Prevention at DFCI (CCGP; Boston, MA; $N = 162$), Tata Medical Center (DGO; Kolkata, India; $N = 20$), Pomeranian Medical University (IHCC; Szczecin, Poland; $N = 52$), and University of Pennsylvania (UPenn; Philadelphia, PA; $N = 132$). Samples from patients with genetically confirmed *BRCA1/2* status were included in the study. Patients with ovarian cancer history or other cancer diagnosed within 1 year from sampling were excluded. Patients with benign adnexal masses were included. Patients with missing diagnoses or *BRCA* status were excluded. All study samples were collected under locally approved institutional review board protocols after obtaining informed consent from study subjects. Sample-level data are presented in Supplementary Data 6. The sequencing methods and NGS panels used for genetic diagnostics of *BRCA* mutations changed over time but the methods used were CLIA-validated (or certified by respective national boards of laboratory diagnostics or genetics in Poland and India) and the geneticists responsible for determining the pathogenicity of *BRCA* mutations adhered to the guidelines of the American College of Clinical Genetics current at the time of testing[50–52].

### Next-generation sequencing

Total RNA was extracted, followed by size-selection, adapter ligation, and library preparation as previously described[32]. All miRNA sequencing data were mapped to the reference miRNA database (miRBase version 22.1) using nf-core/smrnaseq version 1.1.0, a uniform, standardized bioinformatic pipeline developed and published as a part of the Nextflow project[53]. Reads unmapped to miRbase were subsequently mapped to human genome GRCh38. The sequencing protocol was set as QIAseq, Illumina or Nextflex, as appropriate to each sample set (QIAseq miRNA sequencing in BWH, IHCC, DFCI and UPenn; Illumina miRNA sequencing in CCGP and NEXTFLEX small RNA sequencing in DGO). All parameters of the pipeline were kept at the default values recommended by the code authors to assure reproducibility. Raw sequencing data in FASTQ files are deposited in Sequence Read Archive under BioProject number PRJNA898621. Derived expression data are available in Supplementary Data 7 and deposited in Gene Expression Omnibus (GEO) under the accession number GSE226445.

## Data integration and miRNA selection

miRNAs were filtered for species detected in at least 33% of the samples in each group at a minimum detection threshold of >=10 transcripts per million (TPM). After filtering, 227 of the initial 2621 miRNAs were retained. Principal Component Analysis (PCA) was used to visualize the presence of batch effects (Supplementary Fig. 1). After voom normalization and mean-variance trend removal (model formula used for voom: -0 + brcaStatus + havingOvaries + group, Supplementary Dataset 8), ComBat was used to combine data from all subject groups (Supplementary Data 9, with the UPenn group serving as reference (model formula used for ComBat: -brcaStatus + havingOvaries)[54,55]. As different technologies were used to quantify the miRNA content in different subject groups, use of an empirical Bayes framework (ComBat) was a necessary step to combine data from all subject groups while accounting for technical heterogeneity[55]. However, to limit the potential confounding influence of ComBat on the effect of interest, we performed two versions of differential expression analysis (Fig. 1b): with and without batch adjustment and compared the results to identify miRNAs detected in both variants. Differential expression analysis was performed using limma[56]. The model formula for limma included the following effects: *BRCA1/2* mutation and the effect of prior bilateral salpingo-oophorectomy (-0 + brcaStatus + havingOvaries). Visualization of the samples in reduced dimensionality space was performed using uniform manifold approximation projection (UMAP)[57]. The settings were as follows: number of neighbors for representation: 10 for batch-adjusted data and 5 for unadjusted, minimal distance: 0.2 for batch-adjusted data and 0.9 for unadjusted, distance metric: Euclidean in both cases. Hierarchical clustering was performed using the Ward method for linkage, Euclidean distance metric for samples (columns) and correlation distance metric for miRNAs (rows)[58]. DE was performed in R (version 3.6.3) with limma (3.42.2), edgeR (3.28.1), sva (3.34.0) and reticulate (1.26), while results' visualization and part of preprocessing was done in Python (version 3.8.10) with pandas (1.3.0), numpy (1.20.0), sklearn (1.0.2), statsmodels (0.11.1), matplotlib (3.3.0), and seaborn (0.11.1).

## Model development and statistical analysis

In this step, to assure the strict external validation, the dataset was divided into training ($N = 391$, 75% of cases from all groups except UPenn, random split), testing ($N = 130$, 25% of cases from all groups except UPenn, random split) and validation ($N = 132$, only UPenn group) sets. Model development and validation were conducted using in-house OmicSelector software (version 1.0; https://biostat.umed.pl/OmicSelector[59]). Briefly, OmicSelector tests 94 feature selection approaches based on 25 distinct variable selection methods. OmicSelector-based feature selection followed initial consistency-based preselection as described above. Feature sets with more than 10 miRNAs were filtered out. Selected feature sets were ranked using 4 modeling techniques (logistic regression, conditional decision trees, recursive partitioning trees, and artificial neural networks with 1 hidden layer) with hyperparameter optimization (2000 random hyperparameter sets) and hold-out validation on the testing set. The number of modeling techniques was reduced to assure low complexity of resulting models, and thus reduce the chance of overfitting. The best model was chosen based on the highest validation accuracy.

To assess model performance, the training area under the ROC was analyzed, and a cutoff value for *BRCA* status prediction was chosen based on the highest Youden index[60]. This cutoff was applied for prediction on testing and validation sets. Accuracy, sensitivity, specificity, positive predictive value (PPV) and negative predictive value (NPV)[61] were calculated for all sets. Where indicated, the alpha level for statistical significance was set at <0.05. Supplementary Code 1 contains the RDS object containing Caret wrapper for final model.

All analyses were performed in R.

## Statistics and reproducibility

Statistical methods applied in the study are described above. Differential expression analysis can be reproduced using code, data and instructions available at our departmental self-hosted GitLab repository https://git.btm.umed.pl/ZBiMT/brca-mirna and on Zenodo at https://doi.org/10.5281/zenodo.7817763. Code and data for the classification model are available on GitHub at https://github.com/kstawiski/brca-classifier and on Zenodo at https://doi.org/10.5281/zenodo.7817845.

No statistical method was used to predetermine sample size. Inclusion and exclusion criteria are specified in the Samples subsection. *BRCA* status and other available clinical data were known to researchers responsible for feature selection and models development with exception of *BRCA1/2* status in validation set that was unknown to the researcher developing models. Models were developed with the use of results of genetic testing for *BRCA* status; thus, modeling outcomes were unknown when those tests were performed. All hypothesis tests were two-sided. Randomization was not applicable to this clinical study as no clinical intervention was performed.

## Reporting summary

Further information on research design is available in the Nature Portfolio Reporting Summary linked to this article.

## Data availability

Raw sequencing data are publicly available in the Gene Expression Omnibus under accession numbers PRJNA898621 and GSE226445 Source data are provided with this paper.

## Code availability

The code for differential expression analysis was deposited at https://git.btm.umed.pl/ZBiMT/brca-mirna and https://doi.org/10.5281/zenodo.7817763; code for the classification model is available at https://github.com/kstawiski/brca-classifier and https://doi.org/10.5281/zenodo.7817845.

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

## Acknowledgements

This work was supported by the Developmental Research Program of the Dana-Farber/Harvard Cancer Center Ovarian Cancer SPORE (Di.C. and K.E.) from the National Institutes of Health and the National Cancer Institute (grant 1P50CA240243-01A1, Di.C.), the Massachusetts Life Sciences Center Bits to Bytes Program (K.E.), the Deborah and Robert First Family Fund (K.E.), the Honorable Tina Brozman Foundation (K.E. and Di.C.), the V Foundation (Di.C.) and the Mighty Moose Foundation (Di.C.) and the DST-UKIERI grant no DST/INT/UK/P-134/2016 (A.M.). The funding organizations had no role in the design and conduct of the study; collection, management, analysis, and interpretation of the data; preparation, review, or approval of the manuscript; or decision to submit the manuscript for publication.

## Author contributions

Dr. K.E., Dr. W.F., and Dr. Di.C. had full access to all the data in the study and take responsibility for the integrity of the data and the accuracy of the data analysis. Concept and design: K.E., W.F., and Di.C. Acquisition, analysis, or interpretation of the data: J.L., H.S., S.D., D.C.C., Do.C., A.M., J.G., J.K., C.L., W.F., Z.N., K.S., U.S., and J.W. Drafting of the manuscript: W.F., K.S., K.E., Z.N., and U.S. Critical revision of the manuscript: K.E., Di.C., J.C., P.K., J.G., and S.D. Statistical analysis: U.S., K.S., W.F., and J.W. Supervision: K.E., W.F., and Di.C.

## Competing interests

K.E., W.F., K.S., and Di.C. are co-inventors of patent US201762444085P/EP3565903A1 (title "Circulating microrna signatures for ovarian cancer"), which relates to the use of circulating miRNAs for ovarian cancer diagnosis. Dr. Elias, Dr. Fendler, and Dr. Chowdhury acknowledge research funding from Aspira Women's Health. K.E. reports research funding from Abcam, Inc. The remaining authors declare no competing interests.
