## [Peer Review File · Nature Communications]

REVIEWER COMMENTS

Reviewer #1 (Remarks to the Author): expertise in cancer prediction statistical analysis

The manuscript from Elias, et al., introduced a miRNA-based approach to predict BRACness in healthy normal. Using BRAC mutation status as a gold standard, a logistic regression model was determined to achieve the best prediction accuracy. The study included a decent number of participants from relatively balanced BRAC wild-type and mutation classes and split them appropriately into training, testing and validation sets. The consistent model performance across these datasets lends credence to the confidence of the model quality. The batch effect within the data is a major concern, and the effect of batch adjustment is not easy to evaluate. However, I can see the authors have made an effort addressing and disclosing the potential disadvantage. Overall, I found this study well designed and the manuscript well written.

My comments:

1. The authors claimed that the miRNA model performance was evaluated using the cut-off derived based on Youden index. While this cut-off is probably optimal in terms of data science, it is a concern whether this cut-off is suitable for real-world application and how well the miRNA model performs in that condition. The authors did mention that the desired use of this model is "first pass screening", meaning that the prevalence of BRAC mutation in the corresponding population is much lower than those used in model development. I recommend the authors to expand their current discussion and to include an estimation of model performance based on a purpose-specific cut-off.
2. It is a surprise that the logistic regression model outperformed all the other candidate models. I did notice a considerable difference in the participating subject cohorts, which may contribute to the triumph of the logistic regression model. In this case, it is desirable to see evaluation on the contribution from each miRNA (feature importance) in each participating cohort to understand heterogeneity.
3. In supplementary material the authors provided individual prediction results and corresponding clinical data. As a standard model evaluation process, the authors need to evaluate whether there is any association between the mis-predictions and the clinical variables.
4. Please provide more detail on how BRAC1/2 mutation status were determined in the Method section.

Reviewer #2 (Remarks to the Author): expertise in miRNA signature development

The present manuscript entitled- Identification of BRCA1/2 mutation carriers using circulating microRNA profiles, by Kevin Elias and colleagues, investigated the prognostic potential of a ten-microRNA signature in identifying BRCA wild-type and BRCA1/2 mutation status in patients with ovarian and breast cancers. The authors performed a small RNA sequencing in patients and analyzed the prognostic nature using informatic approaches. The authors found that a ten-microRNA signature can be a predictive marker of BRCA1/2 mutation. The study had done in multiple cohorts involving 653 patients, and the results have clinical significance. Therefore, I have the highest enthusiasm. This manuscript should be accepted, providing the following information.

1. The authors should present detailed analysis methods for each figure, including the type of analysis, program/package used in 'R', name of statistical analysis, and the cut-offs used.
- 2) The main goal of this manuscript is not clear. Is this identifying mutant BRCA1/2 in ovarian and breast cancer patients or patients with a post or premenopausal condition? This should reflect in the text of the abstract.
- 3) It is unclear whether patients of the six cohorts belong to either ovarian or breast cancer. This information should be updated in Figure 1A.
- 4) Figure 1B, text within the last box, unmodified should be unmodified. It is a typo error. Figure 1 B's text font should be bigger or equal to Figure 1A.
- 5) In Figure 1C-E, the text font should be bigger. With the given size, it is hard to read.
- 6) Figure 1 legend is incomplete. For example, the legend gives C, D: 2 different colors for miRNAs but not described in the legend. Similarly, Cohort colors did not match in Figures 1A and B.
- 7) ROC analysis numbers are missing in Figure 3A.
- 8) The code used in 'R' should be given in the supplementary data concerning each figure.

Reviewer #3 (Remarks to the Author): expertise in BRCA genomics

The main aim of the study is to identify a panel of miRNAs that can be used to identify BRCA1/2 mutation carriers. The authors have examined the serum microRNA profile of 350 healthy BRCA1/2 mutation carriers and compared them with 303 healthy controls from multiple different hospitals/institutions. A total of 19 miRNAs were first identified that could separate the BRCAmt and BRCAwt groups. Further statistical analysis and modelling resulted in the identification of 10 miRNAs that could distinguish between the two groups with high accuracy, sensitivity, and specificity. One of these is miR-182-5p, which regulates BRCA1 expression.

Most previous studies focused on the identification of BRCA1/2 deficient genomic signatures have been performed in tumor samples. Previous attempts to identify miRNA signatures have been performed in individuals with the tumors. This is the first study where the miRNA profile has been examined in healthy individuals. Some participants (11.5%) had undergone salpingo-oophorectomy because of the presence of BRCA1/2 mutation. If these 10 miRNAs can successfully identify BRCA1/2 mutation carriers in a larger cohort, it will be a major breakthrough and will have a significant impact on genetic screening to identify individuals at risk of developing the disease. My only concern is related to the mechanism of such a distinct miRNA profile in healthy heterozygous carriers and the rationale of combining BRCA1 and BRCA2 mutation carriers. These should be addressed by the authors:

1. Mechanistically, it is hard to comprehend how BRCA1 and BRCA2 mutation carriers exhibit a similar miRNA profile. While loss of both, BRCA1 and BRCA2, renders cells deficient in homologous recombination (HR), their biological functions are very different. Furthermore, HR is not affected in heterozygous cells. Some studies have reported haploinsufficiency of BRCA1 can contribute to a defect in replicative stress response and a defect in replication fork stability. Such defects have not been observed in BRCA2 heterozygous cells. BRCA2 mutant cells exhibit aldehyde sensitivity that leads to replicative stress. It is difficult to understand how these miRNAs are dysregulated in BRCA1/2 het cells. If the miRNA profile reflects the genomic instability acquired by the cells in mutation carriers, this profile should identify individuals with mutations in other breast and ovarian cancer susceptibility genes that play an important role in HR such as PALB2, RAD51C. Can these miRNAs be used to detect BRCAness?

2. The description of the mutations identified in BRCA1/2 mutation carriers is not clear. Are all the mutations predicted to result in complete loss of function, such as those resulting in a truncated protein. Or the mutations also include single amino acid variants, which may or may not result in complete loss of function. How such mutations can have a similar miRNA profile in heterozygous state is difficult to understand.

RESPONSE TO REVIEWERS' COMMENTS

Reviewer #1 (Remarks to the Author): expertise in cancer prediction statistical analysis

The manuscript from Elias, et al., introduced a miRNA-based approach to predict BRACness in healthy normal. Using BRAC mutation status as a gold standard, a logistic regression model was determined to achieve the best prediction accuracy. The study included a decent number of participants from relatively balanced BRAC wild-type and mutation classes and split them appropriately into training, testing and validation sets. The consistent model performance across these datasets lends credence to the confidence of the model quality. The batch effect within the data is a major concern, and the effect of batch adjustment is not easy to evaluate. However, I can see the authors have made an effort addressing and disclosing the potential disadvantage. Overall, I found this study well designed and the manuscript well written.

We would like to thank the reviewer for careful reading of our manuscript and for the positive feedback. Specific comments raised are addressed below.

My comments:

1. The authors claimed that the miRNA model performance was evaluated using the cut-off derived based on Youden index. While this cut-off is probably optimal in terms of data science, it is a concern whether this cut-off is suitable for real-world application and how well the miRNA model performs in that condition. The authors did mention that the desired use of this model is "first pass screening", meaning that the prevalence of BRAC mutation in the corresponding population is much lower than those used in model development. I recommend the authors to expand their current discussion and to include an estimation of model performance based on a purpose-specific cut-off.

We confirm that the rationale behind choosing the Youden index, which maximizes accuracy, was based on a data science approach and the need to evaluate whether the observed signature provides a result as similar as possible in the validation group to that in the training set. The Youden index was thus chosen to establish an optimal cut-off for group separation and allow for the model's performance testing in the validation cohort and we fully agree that it does not have to be the optimum one from the standpoint of clinical utility. For individual-level decisions and practical deployment of the model, having a broader perspective on the performance of the model with varying cut-off values or even just calculating the probability of BRCAmt may be more feasible. To demonstrate this, we added a supplementary figure (3) depicting the impact of different cut-off probability values in the whole study cohort on positive and negative predictive values and added a relevant fragment to the discussion section of the manuscript. For an exemplary cutoff $p=0.25$, in a population with 0.4% prevalence of BRCA1/2 mutations, the negative predictive value would be 99.6% and the positive predictive value would be 8.7%. Nevertheless, a continuous probability score result rather than a dichotomous one may be more clinically useful in clinical management and as such exact probabilities for all patients in the presented group are reported in supplementary table 10. Using the code provided these values can be calculated for any new individuals as well, although given the technical aspects of miRNA-sequencing, the definitive test for clinical application will likely have to be downscaled to a method less expensive and easier to deploy, like qPCR.

2. It is a surprise that the logistic regression model outperformed all the other candidate models. I did notice a considerable difference in the participating subject cohorts, which may contribute to the triumph of the logistic regression model. In this case, it is desirable to see evaluation on the

contribution from each miRNA (feature importance) in each participating cohort to understand heterogeneity.

The purpose of the study was primarily to determine whether circulating microRNAs may be used to gain insight into the efficiency of homologous recombination pathway and thus higher risk of certain malignancies. The repertoire of classification models that could be used to address this issue from a data mining perspective is large and we evaluated multiple methods reliant on different statistical assumptions and techniques (as depicted in Supplementary Table 6). However, as the data presented at this stage are generated through high throughput sequencing, the key issue at this stage was not to focus on the performance of the model itself, but rather on identifying variables with the best potential for class separation through any possible means. Furthermore, that set of variables and the classification models built upon them would have to successfully validate on the UPenn cohort. We observed an abundance of microRNAs that were significantly and reproducibly up or down regulated making a clear class separation fairly easy without the need for complex rules or segmentation. Thus, we believe that the inherent simplicity and robustness of logistical regression were the winning factors here and that the capability of the model to efficiently separate the BRCAmt and BRCAwt classes without overfitting, allowed for very good accuracy in the validation cohort.

More complex methods were evaluated and included artificial neural networks, Bayesian techniques, support vector machines and decision trees, but none of them were significantly better than logistic regression which motivated us to remain with the fairly simple, but efficacious and robust final model as a demonstration of a diagnostic tool. Parameters of the final logistic regression equation were presented in Supplementary table 7. Please note that as the crucial step was to select a subset of miRNAs with good class separation capabilities, the 10 miRNAs that entered the logistic regression model were all significant in univariate comparisons, but some have lost significance on the final multivariable model building stage. Further reduction of the model through stepwise selection would likely be possible, but this would require an additional layer of variable selection which we did not feel sufficiently justified to apply. We are aware, that in order to deploy the proposed test in the clinical setting, it would be advisable to downscale the testing method and use the final, trimmed down set with adequate normalization factors. Development of such a test however, will require a substantially different experimental plan and considering a combinatorial test for HR deficiency and malignancies associated with this status.

3. In supplementary material the authors provided individual prediction results and corresponding clinical data. As a standard model evaluation process, the authors need to evaluate whether there is any association between the mis-predictions and the clinical variables.

We presented individual level probabilities of all patients in supplementary table 10 and show them graphically on Figure 3C and 3D. To better visualize the performance in subgroups we added a Supplementary Table 9 which crosstabulates true and predicted classes in the respective subgroups for menopausal status. Patients after risk reducing salpingo-oophorectomies all had BRCA1 or BRCA2 mutations making it impossible to directly compare sensitivities and specificities with those without/before RRSO. Regarding age – accurate data was provided only for 340 individuals with a predominance of controls, no correlation of predicted probabilities was noted across the whole group and subgroups of BRCAmt and BRCAwt patients – we represented this on Supplementary Figure 4 added to the manuscript. As exact data on age was available predominantly for controls direct estimation of the model's performance may be biased, however, even in this limited sample

there was no evident correlation of age categories with sensitivity, specificity and accuracy of the test – data was presented in Supplementary Table 11.

4. Please provide more detail on how BRCA1/2 mutation status were determined in the Method section.

We added information about the diagnostic methods used by the participating centers to the methods section. The sequencing methods and NGS panels used for genetic diagnostics changed over time but the methods used were CLIA-validated (or certified by respective national boards of laboratory diagnostics or genetics in Poland and India) and the geneticists responsible for determining the pathogenicity of BRCA mutations adhered to the guidelines of the American College of Clinical Genetics.

Reviewer #2 (Remarks to the Author): expertise in miRNA signature development

The present manuscript entitled- Identification of BRCA1/2 mutation carriers using circulating microRNA profiles, by Kevin Elias and colleagues, investigated the prognostic potential of a ten-microRNA signature in identifying BRCA wild-type and BRCA1/2 mutation status in patients with ovarian and breast cancers. The authors performed a small RNA sequencing in patients and analyzed the prognostic nature using informatic approaches. The authors found that a ten-microRNA signature can be a predictive marker of BRCA1/2 mutation. The study had done in multiple cohorts involving 653 patients, and the results have clinical significance. Therefore, I have the highest enthusiasm. This manuscript should be accepted, providing the following information.

1. The authors should present detailed analysis methods for each figure, including the type of analysis, program/package used in 'R', name of statistical analysis, and the cut-offs used.

We added suitable information from the methods section to appear also in the figure legends. The final version of the code used for differential expression analysis and classification model development will be deposited as described in the code availability statement of the paper in the Zenodo repository.

2) The main goal of this manuscript is not clear. Is this identifying mutant BRCA1/2 in ovarian and breast cancer patients or patients with a post or premenopausal condition? This should reflect in the text of the abstract.

The manuscript focused on patients that were screened for BRCA1 or BRCA2 mutations due to clinically-defined high risk. All were malignancy-free at the moment of testing and for at least a year thereafter. We have modified the abstract accordingly to better highlight that fact.

3) It is unclear whether patients of the six cohorts belong to either ovarian or breast cancer. This information should be updated in Figure 1A.

Patients with diagnosed malignancies were excluded from this analysis of clinical data collected at their respective centers through medical examination and imaging studies were relevant. As stated in the methods section, patients with prior malignancy of ovarian or other cancers were excluded as were ones with malignancies diagnosed within one year after sample collection to rule out the

presence of subclinical, undiagnosed disease at the moment of testing. It is possible that some of them would develop malignancies in long term observations according to their inherently increased risk. For the sake of the tested hypothesis however – detecting high risk due to BRCA1/2 mutations, the subsequent development of malignancies would be irrelevant as after identification of high genetically-determined risk, the patients would have entered oncologic surveillance which would facilitate the diagnosis of those malignancies under current best practice.

4) Figure 1B, text within the last box, unmodified should be unmodified. It is a typo error. Figure 1 B's text font should be bigger or equal to Figure 1A. 5) In Figure 1C-E, the text font should be bigger. With the given size, it is hard to read. 6) Figure 1 legend is incomplete. For example, the legend gives C, D: 2 different colors for miRNAs but not described in the legend. Similarly, Cohort colors did not match in Figures 1A and B.

All three figures were corrected accordingly to improve the visibility of data and correct the indicated typos. Legend on figures 2C and D (volcano plots) was provided on the graph; For figures 2A & 2B the markings represent the same cohorts with orange depicting patients without mutations and blue those with either BRCA1 or BRCA2mt. There was no possibility to match the color of 2a/b with 2C and 2D as the first two panels represent biological information whereas color coding on 2c and 2 represents significance, fold change and similarity between 2C and 2D. We added an explanation on this to the legend.

7) ROC analysis numbers are missing in Figure 3A.

We added the information on area under the curve to the figure legend.

8) The code used in 'R' should be given in the supplementary data concerning each figure.

The final version of the code used for differential expression analysis and classification model development will be deposited as described in the code availability statement of the paper in the Zenodo repository.

Reviewer #3 (Remarks to the Author): expertise in BRCA genomics

The main aim of the study is to identify a panel of miRNAs that can be used to identify BRCA1/2 mutation carriers. The authors have examined the serum microRNA profile of 350 healthy BRCA1/2 mutation carriers and compared them with 303 healthy controls from multiple different hospitals/institutions. A total of 19 miRNAs were first identified that could separate the BRCAmt and BRCAwt groups. Further statistical analysis and modelling resulted in the identification of 10 miRNAs that could distinguish between the two group with high accuracy, sensitivity, and specificity. One of these is miR-182-5p, which regulates BRCA1 expression.

Most previous studies focused on the identification of BRCA1/2 deficient genomic signatures have been performed in tumor samples. Previous attempts to identify miRNA signatures have been performed in individuals with the tumors. This is the first study where the miRNA profile has been examined in healthy individuals. Some participants (11.5%) had undergone salpingo-oophorectomy because of the presence of BRCA1/2 mutation. If these 10 miRNAs can successfully identify BRCA1/2 mutation carriers in a larger cohort, it will be a major breakthrough and will have a significant impact on genetic screening to identify individuals at risk of developing the disease. My only concern is related to the mechanism of such a distinct miRNA profile in healthy heterozygous carriers and the

rationale of combining BRCA1 and BRCA2 mutation carriers. These should be addressed by the authors:

1. Mechanistically, it is hard to comprehend how BRCA1 and BRCA2 mutation carriers exhibit a similar miRNA profile. While loss of both, BRCA1 and BRCA2, renders cells deficient in homologous recombination (HR), their biological functions are very different. Furthermore, HR is not affected heterozygous cells. Some studies have reported haploinsufficiency of BRCA1 can contribute to defect in replicative stress response and a defect in replication fork stability. Such defects have not been observed in BRCA2 heterozygous cells. BRCA2 mutant cells exhibit aldehyde sensitivity that leads to replicative stress. It is difficult to understand how these miRNAs are dysregulated in BRCA1/2 het cells.

We agree with the reviewer that while the existing literature supports the existence of diminished HR capacity in both *BRCA1* or *BRCA2* mutant cells, heterozygous mutations are not sufficient to cause HR deficiency in the absence of other stressors. And as the reviewer stated, *BRCA1* haploinsufficiency has been linked to replication stress and defects in fork stability. For *BRCA2*, it has been shown by Venkitaraman and colleagues (Cell, 2017) at the cellular level that transient exposure to natural concentrations of aldehyde induced *BRCA2*-haploinsufficiency via replication stress, suggesting that carcinogenesis in mutation carriers may be instigated by environmental and endogenous compounds. Supporting the link between either *BRCA1/2* mutation and replicative stress, in healthy women with *BRCA1* or *BRCA2* haploinsufficiency there are significantly increased immune cell infiltrates in breast tissue (Ogony et al, 2022) and in serum of *BRCA1* and 2 mutation carriers there is increased thymidine kinase activity and soluble EGFR (sEGFR) (Nisman et al, 2013). Having said that, we completely agree with the reviewer that *BRCA1* and *BRCA2* have distinct functions in DNA repair. However at the organismal level, through genome instability or other mechanisms, they have a systemic impact that is reflected in enhanced immune infiltration in breast tissue or enhanced activity of growth factors (TK1 and sEGFR) in serum. Here, we systematically capture *BRCA* haploinsufficiency using serum miRNAs, but the underlying molecular mechanism remains to be determined in future studies. We have discussed these points in the revised manuscript.

If the miRNA profile reflects the genomic instability acquired by the cells in mutation carriers, this profile should identify individuals with mutation in other breast and ovarian cancer susceptibility genes that play an important role in HR such as PALB2, RAD51C. Can these miRNAs be used to detect BRCAness?

We would like to thank the reviewer for this excellent comment. To evaluate this hypothesis, at least preliminarily, we performed sequencing in 7 available patients (5 from the DFCI biobank cohort, 2 from the UPenn) with mutations of *PALB2* (N=6) or *BRIP1* (N=1). Data was processed using the same pipeline and adjustment for cohort of origin. Probabilities of *BRCAMt* are as depicted on the figure below. In all cases, the predicted probabilities were high and in 5/7 would be above the cut-off of 0.58 probability (used as threshold of the model at validation stage) and so would correctly identify these individuals for genetic testing. Two missed samples had a probabilities of 0.35 and 0.52 making the risk of HR haploinsufficiency substantially higher than that of the general population.

Figure – Probabilities of BRCAmt status in individuals with BRCAmt (Cases), controls and 7 individuals with mutations of genes other than *BRCA1* or *BRCA2*.

Thus, the results appear to support our hypothesis that the model will be effective in diagnosing HR gene haploinsufficiency caused by genetic factors other than mutations of *BRCA1* or *BRCA2*. However, as two of the patients with *PALB2* mutations (including one with the lowest calculated probability score of HR haploinsufficiency) had their samples collected during RRSO and had tumors *in situ* undiagnosed clinically and the patient with *BRIP1* was a single individual, the number of patients with nonBRCA HR deficiencies in line with the initial inclusion criteria was extremely limited and thus we feel that this group is too scarce to make a definitive conclusion upon the universality of the test's accuracy and would rather leave this result solely as part of this response rather than of the manuscript. Nonetheless, we would like to thank the reviewer for this excellent comment and we intend to further evaluate this miRNA profile in larger cohorts of patients with non BRCA HR gene haploinsufficiency's. If true, this miRNA profile may even be helpful to identify haploinsufficiency's in high risk-families that are negative for known HR pathway genes based on current genetic testing.

2. The description of the mutation identified in BRCA1/2 mutation carriers is not clear. Are all the mutations predicted to result in complete loss of function, such as those resulting in a truncated protein. Or the mutations also include single amino acid variants, which may nor may not result in complete loss of function. How such mutations can have a similar miRNA profile in heterozygous state is difficult to understand.

Assuring uniformity of diagnosis across all cohorts was no small task so when issuing the call for samples we required the centers to evaluate the patients using the criteria of the American College of Medical Genetics Guidelines. As accurate data on specific mutations was assumed to be difficult to obtain at individual level due to data restriction policies and institutional regulations, we assured that patients with variants of uncertain significance were not included in the cohorts making those designated as BRCAwt negative in at least sequencing of *BRCA1* and *BRCA2*. Any mutations determined as possibly pathogenic or probably pathogenic in genes associated with HR pathway were excluded from the initial dataset presented in the manuscript. Within the limited cohorts of

DFCI, BWH and CCGP we revisited the records where possible to obtain individual level data on types of mutations. No differences were observed in the performance of the model depending on the type of mutations – patients with nonsense, frameshift and missense mutations were assigned non-significantly different probabilities of BRCAmt.

REVIEWERS' COMMENTS

Reviewer #1 (Remarks to the Author):

The authors has now included a good level of discussion on the utility of the proposed miRNA assays in a real-world clinical setting. Though further evaluation is needed on whether miRNA is a feasible tool for BRACness prediction due to its moderate classification accuracy, the information provided now is sufficient as a starting point.

The authors provided more detail on how they train their models, how the best model is selected, and emphasised that the purpose of the paper is to demonstrate the use of miRNA model. While I am still not convinced that the modelling procedure is optimal, I agree with the authors that finding the best data model is not the objective of this paper, and that the reported logistic model is sufficient and suitable to demonstrate the use of miRNA to predict BRACness in a real-world setting.

The authors added in sufficient detail on model performance. I noticed, though, the sTable 7 has not been mentioned anywhere in the main text. Please correct.

Reviewer #2 (Remarks to the Author):

The authors revised the manuscript per the reviewer's comments. Now, this artilo looks much better.

Reviewer #3 (Remarks to the Author):

The authors have satisfactorily addressed my concerns and made appropriate changes in the manuscript.

RESPONSE TO REVIEWERS' COMMENTS

Reviewer #1 (Remarks to the Author):

The authors has now included a good level of discussion on the utility of the proposed miRNA assays in a real-world clinical setting. Though further evaluation is needed on whether miRNA is a feasible tool for BRACness prediction due to its moderate classification accuracy, the information provided now is sufficient as a starting point.

The authors provided more detail on how they train their models, how the best model is selected, and emphasised that the purpose of the paper is to demonstrate the use of miRNA model. While I am still not convinced that the modelling procedure is optimal, I agree with the authors that finding the best data model is not the objective of this paper, and that the reported logistic model is sufficient and suitable to demonstrate the use of miRNA to predict BRACness in a real-world setting.

We greatly appreciate the Reviewer's positive evaluation and are happy that, despite the limitations, our work advances the field of BRCA research and its clinical importance.

The authors added in sufficient detail on model performance. I noticed, though, the sTable 7 has not been mentioned anywhere in the main text. Please correct.

We have added the missing line referencing supplementary table 7 and apologize for this omission.

Reviewer #2 (Remarks to the Author):

The authors revised the manuscript per the reviewer's comments. Now, this artilo looks much better.

We greatly appreciate the Reviewer's response.

Reviewer #3 (Remarks to the Author):

The authors have satisfactorily addressed my concerns and made appropriate changes in the manuscript.

We greatly appreciate the Reviewer's response.